# *Interleukin 28B* Polymorphism as a Predictor of Sustained Virological Response to Sofosbuvir-Based Therapy for Hepatitis C Virus Patients

**DOI:** 10.3390/tropicalmed7090230

**Published:** 2022-09-05

**Authors:** Seham Mahrous Zaki, Hanan Samir Ahmed, Monkez Motieh Yousif, Eman Mohamed Awad

**Affiliations:** 1Clinical Pathology Department, Faculty of Medicine, Zagazig University, Zagazig 44519, Egypt; 2Internal Medicine Department, Faculty of Medicine, Zagazig University, Zagazig 44519, Egypt

**Keywords:** direct-acting antivirals, hepatitis C virus, sofosbuvir, daclatasvir, ribavirin, *interleukin 28B* polymorphism

## Abstract

In various genome-wide correlation studies, interleukin *(IL)28B* gene polymorphism has been strongly correlated with both the therapeutic and spontaneous mediated clearance of hepatitis C virus (HCV). Therefore, this study aimed to evaluate the genotype and allele frequency distributions of *IL28B (rs12979860)* in patients with chronic hepatitis C and assess the *IL28B* polymorphisms as predictors of sustained virological response to SOF-based therapy for HCV in Egyptian patients. This retrospective case-control study was conducted on 54 chronic HCV patients who completed treatment with SOF/DCV ± RBV for 12 weeks and responded to treatment with SVR12 (the responder group) as a control group, and 54 chronic HCV patients who completed treatment with SOF/DCV ± RBV for 12 weeks and did not respond to treatment and failed to achieve SVR12 (the non-responder group) as a case group. The CC genotype frequency of *IL-28B (rs12979860)* was greater in the responder group (51.9%). In contrast, the TT genotype frequency was higher in the non-responder group (48.1%) (*p* < 0.001), and the T allele significantly increased the risk of non-responses by 3.13 fold. Therefore *IL-28B (rs12979860)* SNP could be used as a genetic predictor of sustained virological response to SOF+DCV ± RBV-based HCV treatment in Egyptian patients.

## 1. Introduction

Hepatitis C virus (HCV) is an increasingly global issue, owing to its significantly increased mortality and morbidity. The global burden of HCV-related diseases, including hepatocellular carcinoma (HCC), cirrhosis, liver transplantation, and liver-associated mortality, increases as infected patients progress to late-stage liver infections [1].

In 2015, 71 million individuals were infected with chronic HCV, with a global incidence of 1.0% of the total population. The Eastern Mediterranean Region had the greatest population incidence (2.3%), accompanied by the European Region (1.5%) [2].

The World Health Organization (WHO) reported that approximately 399,000 individuals died from hepatitis C in 2016, most of which died from cirrhosis and hepatocellular carcinoma (HCC) [3].

In 2015, the epidemiological statistics from the Egypt Demographic and Health Survey indicated that the HCV antibody prevalence was 10.0% and that of the HCV RNA was 7.0% in the 15–59-year age groups, and approximately 3.7 million people had chronic HCV disease [4].

In Egypt, 92.5% of patients were infected with HCV genotype 4, 3.6% with genotype 1, 3.2% with multiple genotypes, and <1% with other genotypes [1,5,6].

With direct antiviral agent (DAA)’s availability, Egypt established a national HCV therapy program in 2014 to treat 250,000 patients chronically infected with HCV yearly, in order to accomplish a national chronic infection incidence of <2% by 2025 [7].

Sofosbuvir (SOF) is an NS5B polymerase nucleotide inhibitor (NI) with a substantial obstacle to resistance and antiviral action against all genotypes that have demonstrated clinical efficacy in treating chronic hepatitis C infection in genotypes 1–6 when used in combination with other DAAs [8].

Sofosbuvir and daclatasvir ± ribavirin treatment combination has proven to be safe and effective in several studies in Egypt, with an overall SVR12 rate above 90% achieved. The predictive factors associated with lower SVR 12 were the presence of cirrhosis, especially Child-Pugh class B, C, older age, and low platelet count. In contrast, the addition of RBV to treatment was associated with a significant increase in the SVR12 rate [9,10,11].

Four *IFN λ* proteins *(IFN λ 1 (IL29), IFN λ 2 (IL28A), IFN λ 3 (IL28B), and IFN λ 4)* are expressed by human genes. Human *IFN λ 1, IFN λ 2, and IFN λ 3* were detected in 2003, whereas *IFN λ 4* was recognized via primary human hepatocyte RNA sequencing in 2013 [12].

In clinical trials the non–CC *IL28B* genotype has been strongly associated with reduced response, and sustained virologic response occurred in 93 out of 95 patients (98%) with the CC genotype of *IL28B*, compared to 202 out of 232 patients (87%) with the non–CC *IL28B* genotype [13].

Previous findings of the treatment response demonstrated that the CC (main) genotype of *IFN λ 3 (rs12979860)* has been correlated with the possibility of attaining a better sustained virological response (SVR) to the combination of daclatasvir (DCV) and SOF treatment in genotype-3-infected HCV [14].

Patients with the *IFNL4 rs12979860 (IL28B) TT* genotype have been ~4.5 times more likely to relapse than those with *rs12979860-CC,* with relapse rates of 0–2% for *rs12979860-CC* genotype compared to 9–10% for *rs12979860-TT* [15].

In the POLARIS-2 clinical trial, following this pan-genotypic treatment (sofosbuvir/velpatasvir/voxilaprevir), the patients with *IFNL4 rs12979860* genotype might relapse. Only 1.2% of patients with *IFNL4*
*rs12979860**-CC* genotype relapsed, compared to 5.9% of patients with *IFNL4 rs12979860-CT* and 4.9% of patients with *IFNL4 rs12979860-TT* genotype. Patients with an unfavorable *IFNL4 rs12979860* genotype (CT or TT) were almost five times more likely to relapse than those with the *IFNL4 rs12979860-CC* genotype [16].

*Interleukin 28B* polymorphism as a predictor of sustained virological response to sofosbuvir-based therapy for HCV as a part of sofosbuvir efficacy is more investigated in clinical trials and on HCV genotypes 1, 2, and 3, while its real-life investigation studies particularly in Egypt on HCV genotype 4 is insufficient [10,17,18] As a result pretreatment *interleukin 28B* genotyping can still be valuable in determining patients who may benefit from shorter treatment courses or deciding the most appropriate treatment decisions when the resources are limited or, in challenging conditions such as the high prevalence as in Egypt with evidence for some continuous HCV transmission [5].

Therefore, this real-life case-control retrospective study aimed to evaluate the genotype and allele frequency distributions of *IL28B (rs12979860)* in Egyptian patients with chronic hepatitis C disease who obtained *SOF*/DCV ± ribavirin (RBV) and assess *IL28B* polymorphisms as predictors of SVR to SOF based therapy for HCV in Egyptian patients, which is essential to provide more practical information for better management of chronic hepatitis C patients.

## 2. Materials and Methods

### 2.1. Study Population

The Clinical Pathology Department of the Faculty of Medicine at Zagazig University and the Zagazig Viral Hepatitis Treatment Center in Sharkia Governorate collaborated in this case-control retrospective study.

Ethical consideration: The Zagazig University Institutional Review Board (ZU-IRB) provided ethical approval for this study (approval no. 3734-14-5-2017). The sample collection started in November 2018 and ended in April 2019. Informed written consent was gathered from all patients to utilize their specimens and clinical information in this study and to publish this paper in accordance with the Helsinki Declaration.

The sample size was calculated using the Epi Info program 6 (Atlanta, GA, USA). By considering odds ratio (OR = 7.9) and expected proportion in controls = 0.8 from previous studies [13] and at the power of study of 80% and a confidence interval (C.I) 95%, the total sample size was calculated to be 108 patients.

This retrospective case-control study was conducted on 54 chronic HCV patients who completed treatment with SOF/DCV ± RBV for 12 weeks and responded to treatment with SVR12 (the responder group) as a control group, and 54 chronic HCV patients who completed treatment with SOF/DCV ± RBV for 12 weeks and did not respond to treatment and failed to achieve SVR12 (the non-responder group) as a case group (Figure 1).

### 2.2. Inclusion Criteria

The selected patients fulfilled the following criteria:Adults aged ≥ 18 years Both sexesChronic HCV patients who completed treatment with SOF/DCV ± RBV for 12 weeks and responded to treatment with SVR12Chronic HCV patients who completed treatment with SOF/DCV ± RBV for 12 weeks and did not respond to treatment and failed to achieve SVR12

### 2.3. Exclusion Criteria

Patients co-infected with HBV Patients with HCC as evidenced by alpha fetoprotein or tissue biopsy Patients who refused to participate in the research

All individuals underwent a full medical history, clinical data acquisition, clinical testing, and anthropometric tests including body weight and height. The body mass index (BMI) was estimated as weight (kg) categorized by height (m) squared, abdominal ultrasonography, laboratory investigations (complete blood count [CBC] by “Sysmex XS” (Sysmex Corporation, Kobe, Japan), liver and kidney functions tests, alpha-fetoprotein (AFP), hepatitis B surface antigen (HBsAg), and hepatitis C antibody (HCVAb) using “Roche Cobas 8000” (Roche Diagnostics, Mannheim, Germany), international normalized ratio (INR) using Sysmex CS 2100 coagulometer (Sysmex Corporation, Japan), HCV RNA quantification using Cobas TaqMan Ampliprep (Roche Diagnostics, Germany) with a lower detection restriction of 15 IU/mL and assessment of liver fibrosis severity using liver ultrasound and FIB-4 = [age (years) × AST (IU/L)]/[Platelet count × 109 × (ALT1/2)].

### 2.4. Treatment Protocol

The patients received treatment as recommended by the National Committee for Control of Viral Hepatitis with both SOF and DCV orally administered at 400 mg/day and 60 mg/day, respectively. RBV was administered for those categorized as not easy to treat; tablets were orally administered in the morning and evening depending on their tolerability and weight (initial dose, 600 mg/day up to 1200 mg/day). The therapy continued for 12 weeks, and the assessment of treatment response was confirmed by HCV RNA PCR at least 12 weeks after completing the course of medication. The treatment responders were labeled as having SVR12 in patients with HCV RNA < 15 IU/mL (undetectable). A measurable HCV RNA ≥ 15 IU/mL level indicated failure of response to treatment.

### 2.5. Sampling

Approximately 5 mL of venous blood specimens were obtained from each patient under whole aseptic conditions. The specimens were categorized into 1.0 mL on sterile ethylenediaminetetraacetic acid (EDTA) vacutainer tubes for complete CBC, 2 mL on sterile plain vacutainer tubes for liver and kidney functions tests, AFP, HBsAg, HCVAb, Quantitative PCR for HCV RNA, and 2.0 mL on sterile EDTA vacutainer tubes for *IL-28B rs12979860* polymorphism using real-time polymerase chain reaction (PCR).

*Interleukin 28B (IL28B) rs12979860* single nucleotide polymorphism (SNP) genotyping was carried out by real-time PCR employing TaqMan^®^ SNP Genotyping Assays (Applied Biosystems, Foster City, CA, USA) using the following steps:

### 2.6. DNA extraction

DNA extraction by blood genomic DNA extraction protocol utilizing Thermo Scientific GeneJET genomic DNA purification Kit(ThermoFisher Scientific Inc., Waltham, MA, USA). And the genomic DNA concentration in specimens was quantified utilizing a Qubit Fluorometer (ThermoFisher Scientific Inc., Waltham, MA, USA).

#### Procedure

200 μL of lysis solution and 20 μL of proteinase k solution was added to 200 μL of whole blood and mixed thoroughly by vortexing to obtain a uniform suspension. The suspension was incubated at 56 °C while vortexing occasionally at a thermomixer until the cells were completely lysed (10 min). 200 μL of ethanol (96–100%) was added and mixed by pipetting or vortexing. The prepared lysate was transferred to a GeneJET Genomic DNA Purification Column inserted in a collection tube. The column was centrifuged for 1 min at 6000× *g*. The collection tube containing the flow-through solution was discarded. The GeneJET Genomic DNA Purification Column was placed into a new 2 mL collection tube. 500 μL of Wash Buffer I (with ethanol added) was added and centrifuged for 1 min at 8000× *g*. The flow-through was discarded and the purification column was placed back into the new collection tube. 500 μL of Wash Buffer II (with ethanol added) was added to the GeneJET Genomic DNA Purification Column and centrifuged for 3 min at maximum speed (≥12,000× *g*). The collection tube containing the flow-through solution was discarded, and The GeneJET Genomic DNA Purification Column was transferred to a sterile 1.5 mL microcentrifuge tube. 50 μL of elution buffer was added to the center of the GeneJET Genomic DNA Purification Column Membrane to elute genomic DNA, incubated for 5 min at room temperature, and centrifuged for 1 min at 8000× *g*. The purification column was discarded. The purified DNA was stored at −20 °C.

### 2.7. Amplification and Detection of IL28B rs12979860 SNP Genotyping by Real-Time PCR Utilizing TaqMan^®^ SNP Genotyping Assays (Applied Biosystems, USA)

#### Procedure

2 µL of 20× working stock of TaqMan^®^ SNP Genotyping Assays (which contained sequence-particular forward and reverse primers for the polymorphic *interleukin 28B (IL28B) rs12979860* sequence to be amplified, two TaqMan^®^ MGB probes: One probe labeled with VIC^®^ dye to detect allele 1 (C allele) sequence, and the other labeled with FAM™ dye to detect allele 2 (T allele) sequence) was added to 12.5-µL TaqMan^®^ Genotyping Master Mix (Applied Biosystems, USA) into a sterile microcentrifuge tube. 3.5-µL DNase-free water was then added to each well of the DNA reaction plate, and 7-µL of extracted DNA was pipetted to make the total volume of the reaction mix 25 µL in each well. Real-time PCR was performed on the “Stratagene Mx3005P” platform (Agilent Technologies, Santa Clara, CA, USA) utilizing the following cycling circumstances: initial activation step, 15 min at 95 °C to activate HotStar Taq DNA Polymerase, then 50 cycles of denaturation at 95 °C for 30-s annealing at 55 °C for 1 min and extension at 72 °C for 1 min. The allelic discrimination plate was read and analyzed when VIC-dye fluorescence only was the homozygote for the C allele, FAM-dye fluorescence only was homozygote for the T allele, and both VIC- and FAM-dye fluorescence alleles C and T were heterozygotes (CT).

### 2.8. Statistical Analysis

The software Statistical Package for the Social Sciences version 20 was used for data analysis (IBM SPSS, Version 20.0. Armonk, NY, USA). The means and standard deviations were used to depict quantitative variables, and a chi-square test was used for categorical variables comparison. For ordinal binary data, chi-square for the trend test was utilized. Levene and Kolmogorov–Smirnov assays were used to confirm suppositions in parametric tests. The independent sample *t*-test (for normally distributed data) and Mann–Whitney test (for not normally distributed data) were employed to compare quantitative data between the two groups. To compare quantitative variables between more than two groups, the one-way ANOVA test (for normally distributed data) and the Kruskal–Wallis test (for not normally distributed data) were utilized. A post hoc test and pairwise comparison test were used to measure differences between the two individual groups, with *p*-values < 0.05. Binary logistic regression analysis calculated the odds of factors correlated with non-response. A *p*-value of < 0.05 was selected as the statistical significance level. *p* ≤ 0.001 was considered highly significant.

## 3. Results

Demographics and laboratory variables of the studied groups are described in Table 1. The mean ages of groups 1 (responder group) and group 2 (non-responder group) were 52.39 ± 8.44 years and 48.7 ± 12.82 years, and no statistically significant difference was observed between the responder and non-responder groups regarding age (*p* = 0.081), gender (*p* = 0.178), BMI (*p* = 0.079), and PCR level of HCV RNA before treatment (*p* = 0.077) (Table 1).

Regarding laboratory variables of the studied groups, highly statistically significant variations were observed between groups 1 (responder group) and 2 (non-responder group) concerning ALT (*p* < 0.001), AST (*p* < 0.001), and AFP (*p* = 0.001). Moreover, a statistically significant difference was observed in the total bilirubin (*p* = 0.023) and creatinine (*p* = 0.026) levels. No statistically significant difference was observed in WBCs, HB, PLT, INR, or albumin (Table 1).

As for ultrasound (US) findings of the liver and FIB4 among studied groups, a statistically significant difference was observed between groups 1 and 2 (*p* < 0.001, 0.041, respectively). The percentage of the cirrhotic liver in the non-responder group was 40.7% compared to 7.4% in the responder group by ultrasound and 42.6% in the non-responder group compared to 24.1% in the responder group by FIB4 (Table 2).

There was no statistically significant difference between groups 1 and 2 concerning the therapeutic protocol for SOF/DCV/RBV or SOF/DCV (*p* = 0.054) (Table 2).

Regarding the frequency of *IL-28B (rs12979860)* SNP genotype, the CC genotype frequency of *IL-28B (rs12979860)* was higher in the responder group (51.9%), whereas the incidence of TT genotype frequency was higher in the non-responder group (48.1%) (*p* < 0.001). Moreover, the C allele was more prevalent in the responder group (63%), whereas the T allele was more prevalent in the non-responder group (64.8%) (*p* < 0.001). The TT genotype also significantly increased the risk of the non-response by 5.2 fold. Also, the CT genotype significantly increased that risk by 4.2 fold, and the T allele significantly increased it by 3.13 fold (Table 3).

A statistically significant correlation was found between *IL-28B (rs12979860)* SNP and age (*p* = 0.013), overall bilirubin (*p* = 0.008), AFP (*p* = 0.002), creatinine (*p* = 0. 049), and PCR level of HCV RNA before treatment (*p* = 0.003) (Table 4).

A high statistically significant relation was determined between *IL28B (rs12979860)* SNP and ultrasound findings of the liver (only 5.3% of the CC genotype had cirrhotic liver compared to 37.5% of the TT genotype) (*p* < 0.001) and FIB-4 (patients with the CC genotype had low FIB-4) (*p* = 0.013) (Table 5).

Furthermore, a high statistically significant relationship was found between *IL28B (rs12979860)* and treatment protocol (15.8% of CC genotypes required SOF+DCV+RBV triple therapy compared to 62.5% of the TT genotype) (*p* < 0.001) (Table 5).

Binary logistic regression analysis of parameters correlated with non-response showed that higher ALT levels increased the risk of non-response by 1.044 fold. CT and TT genotypes significantly increased the risk of non-response by 8.768 and 10.364 fold independently. In addition, cirrhosis increased the risk of non-response by 2.412-fold, whereas the presence of parenchymatous liver disease was protective (Table 6).

## 4. Discussion

*The IL-28B* genotype SNP *rs12979860* is located 3 kb upstream of the *IFN 3 λ (IL-28B)* gene. Patients with the *IL28B C/C* allele are two to five times more likely than those with the *IL28B T/T* variant to achieve viral clearance with Peg-IFNα plus RBV therapy [19].

SOF (Sovaldi), a nucleotide analog inhibitor of the HCV NS5B polymerase effective against all six main genotypes, was the first direct-acting antiviral in its category to get the US Food and Drug Administration approval for the treatment of chronic HCV infection in December 2013 [20].

In the era of oral interferon-free therapy treatments combining several DAAs with SVR rates of >90%, the impact of *IL28B* SNP has not been clear yet. [21] Therefore, it might be valuable to investigate *IL-28B (rs12979860)* SNP as predictors of response to SOF/DCV ± RBV treatment.

Regarding the frequency of *IL-28B (rs12979860)* SNP genotype, the CC genotype frequency of *IL-28B (rs12979860)* was higher in the responder group (51.9%), whereas the incidence of TT genotype frequency was higher in the non-responder group (48.1%) (*p* < 0.001). Moreover, the C allele was more prevalent in the responder group (63%), whereas the T allele was more prevalent in the non-responder group (64.8%) (*p* < 0.001). TT genotype significantly increased the risk of non-response by 5.2 fold, whereas the CT genotype significantly increased the risk by 4.2 fold, and the T allele significantly increased the risk of non-response by 3.13 fold. Besides, the binary logistic regression analysis of parameters correlated with non-response showed that CT and TT genotypes significantly increased the risk of non-response by 8.768- and 10.364-fold independently. Therefore, the higher SVR in *IL28B* CC genotype could support their selection for minimal drug combination for a shorter duration with better response and fewer adverse effects, while CT and TT genotypes with increased risk of non-response were treated directly with a pan- genotypic regimen.

The current results were consistent with O’Brien et al. (2014), who analyzed published ION-3 data and showed that SVR ratios significantly differed by *rs12979860 (“IL28B”)* genotype (*p* = 0.03), comprising >98% in rs12979860-CC individuals [22].

Similarly, in the BOSON study, individuals with IL28B CC genotype had significantly higher SVR rates (16-week arm, *p*-value = 0.0015; CC SVR rate, 76% (26 of 34); non-CC SVR rate, 37% (13 of 35); 24-week arm, *p*-value = 0.009; CC SVR rate, 96% (30 of 31); non-CC SVR rate, 73% (29 of 40) [23].

The current findings were consistent with those of Khan et al. (2019). They revealed that the CC genotype of *IL-28B (rs12979860)* was observed at a significantly higher frequency in patients with SVR (39.5%) compared to relapses (10.5%) and non-responders (17%), demonstrating the correlation of CC genotype with the probability of attaining SVR in contrast to non-CC variants. Moreover, the C allele was more prevalent in patients with SVR (65%), whereas the T allele was more prevalent in non-responders (62%) [14].

In addition, the current results followed the phase 2, open-label ACCORDION clinical trial in which SVR rates in those with *IL-28B CC* genotype (91.3%, (21/23)) were more significant than those with IL28B CT (75%, (12/16)) or TT genotype (83.3%, (30/36)) [24].

The findings were inconsistent with the phase III study ALLY-3 clinical trial, which demonstrated that the *IL-28B* genotype did not affect virological consequences [25]. These differences might correlate with differences in the studied population and HCV genotypes.

The present findings disagreed with those of Jacobson et al. (2013), who found that SVR in various patient subgroups did not differ significantly regarding *IL28B*: CC vs. non-CC (*p* = 0.81) [26]. These differences might be associated with differences in the studied population and treatment protocols.

Regarding the relation between *IL28B (rs12979860)* genotype, demographics, and laboratory parameters, this study showed no statistically significant relation between *IL28B (rs12979860)* SNP and BMI, WBCs, hemoglobin, PLT, INR, albumin, ALT, or AST. However, a statistically significant relationship was observed between *IL28B (rs12979860)* SNP and age (*p* = 0.013), overall bilirubin (*p* = 0.008), AFP (*p* = 0.002), and creatinine (*p* = 0. 049). This study showed a statistically significant association between the *IL28B (rs12979860)* genotype and PCR level of HCV RNA before treatment (individuals with CC genotype had lower PCR levels) (*p* = 0.003).

The present results were confirmed by Jordovic et al. (2019), who reported that individuals with non-CC genotypes had more significant baseline viremia, with HCV RNA burdens exceeding 800,000 IU/mL, compared to those with CC genotypes (*p* = 0.014) [27].

Conversely, Nosott et al. (2016) found that the association between pretreatment basal viremia levels and *IL28B* genetic polymorphism *(rs12979860)* distribution was statistically negligible [28].

This study found a high statistically significant difference between responders and non-responders regarding the percentage of the cirrhotic liver. It was 7.4% compared to 40.7%, respectively. Binary logistic regression analysis of factors associated with non-response showed that cirrhosis increased the risk of non-response by 2.412 fold. Also, there was a highly statistically significant relationship between the *IL28B (rs12979860)* genotype and ultrasound findings of the liver (only 5.3% of the CC genotype had cirrhotic liver compared to 37.5% of TT genotype (*p* < 0.001) and FIB-4 (patients with CC genotype had lower FIB-4) (*p* = 0.013). This finding could be valuable because the patient with TT genotype and cirrhosis had the priority for treatment despite significantly lower SVR as also recommended by El-Khayat et al. [11], and *IL28B (rs12979860) SNPs* could be used as a genetic predictor to assess the risk of fibrosis progression in chronic HCV patients. The identification of genetic host factors participating in the pathogenesis and clinical course of HCV immunity is essential to understand disease mechanisms and identifying biological targets for personalized treatments and vaccines, which remain a public health priority as the extensive use of highly effective direct-acting antivirals is unlikely to eradicate HCV without vaccines to limit transmission.

The current results agreed with those of Jordovic et al. (2019), who found that patients with non-CC genotypes had more significant histological action of hepatitis (69.2% vs. 24.1%) and cirrhosis (53.8% vs. 24.1%) than individuals with CC *rs12979860* genotypes (*p* < 0.05) [27].

Furthermore, the present results were consistent with those of Nosott et al. (2016). They reported a correlation between liver fibrosis and *IL-28B* genotype distribution (*p* < 0.05). Individuals with the CT-TT genotype had a more significant percentage of individuals with liver fibrosis levels of >8.7 Kpa (23.2%). [28]

Also, the findings were consistent with those of Mangia et al. (2016). They found a significant association between the frequency of *IL-28B* T alleles and the likelihood of transition to cirrhosis in G1-infected individuals (as detected by biopsy or non-invasive examinations) (*p* = 0.0018) [29].

In contrast, the present results disagreed with those of Rembeck and Lagging (2015), who demonstrated that in comparison to CT and TT variants, the CC variant of *rs12979860* was significantly correlated with a higher ALT and AST to platelet ratio index score, demonstrating a more significant inflammation and fibrosis degree. These differences might be associated with the differences in the studied group and the HCV genotype [30].

This study showed a high statistically significant relationship between the *IL-28B (rs12979860)* genotype and treatment protocol (15.8% of CC genotype required SOF + DCV + RBV triple therapy compared to 62.5% of TT genotype) (*p* < 0.001). This finding could be valuable in selecting individualized treatment protocols for patients with the CC *IL28B* genotype for appropriate drugs combination without ribavirin to limit costs and avoid possible side effects and be utilized as global strategies to optimize treatment resource guidelines at the population level.

## 5. Conclusions

*IL-28B (rs12979860)* SNP could be used as a genetic indicator to determine the degree of fibrosis in individuals with HCV. Moreover, it could be used as a predictor of SVR to SOF + DCV ± RBV treatment for HCV in Egyptian patients. It is useful for choosing an appropriate drug combination to limit additional costs and potential adverse impacts. Identifying *IL28B* SNP as a genetic host parameter involved in the HCV disease pathogenesis is highly recommended to understand the disease mechanisms and detect biological targets for vaccines to achieve HCV eradication.

## 6. The Limitations of the Study

The limitations of this study were its retrospective design, single-center sample collection with a small number of individuals involved, and the absence of data about HCV genotyping and resistance.

## Figures and Tables

**Figure 1 tropicalmed-07-00230-f001:**
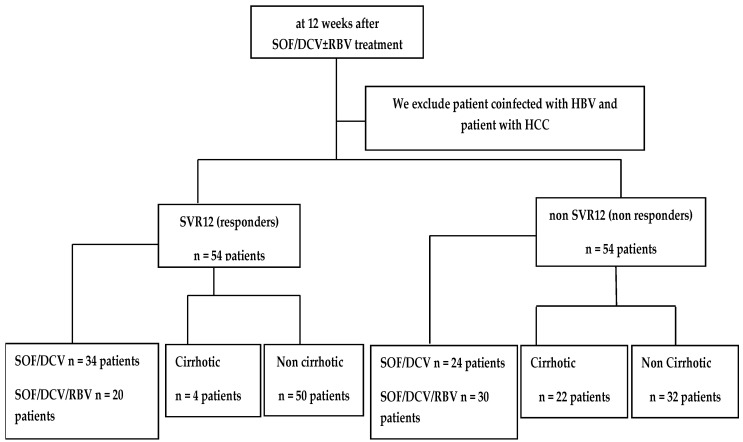
The patients enrolled in the study through the clinical.

**Table 1 tropicalmed-07-00230-t001:** Demographics and laboratory variable characteristics of the studied groups.

	Responders Group I *n* = 54 (%)		Non-Responders Group II *n*= 54 (%)		Test of Significance	*p*
Age (years)						
X ± SD	52.39 ± 8.44		48.7 ± 12.82		*t* test = 1.765	0.081
Range	33–67		21–67			
Gender						
Male	23	(42.6%)	30	(55.6%)	χ^2^ = 1.815	0.178
Female	31	(57.4%)	24	(44.4%)		
BMI (kg/m^2^)						
X ± SD	30.13 ± 2.91		31.83 ± 6.39		*t* test = −1.778	0.079
Range	21.7–34.5		24.5–47.8			
WBCs (10^3^/cmm)						
X ± SD	6.84 ± 1.69		6.17 ± 1.9		1.937	0.055
Range	4.2–10.4		3.4–10.4			
Haemoglobin (g/dL)						
X ± SD	13.49 ± 1.57		13.23 ± 1.54		0.897	0.372
Range	10–17.3		10.8–16.2			
PLT (10^3^/cmm)						
X ± SD	190.15 ± 42.47		196.44 ± 88.19		−0.437	0.638
Range	99–307		57–395			
INR						
X ± SD	1.16 ± 0.22		1.14 ± 0.15		0.513	0.609
Range	1–1.8		1–1.65			
Total bilirubin (mg/dL)						
X ± SD	0.77 ± 0.22		0.89 ± 0.27		−2.307	0.023 *
Range	0.4–1.3		0.3–1.6			
ALT(IU/L)						
Median	37		45.1		−3.566	<0.001 **
Range	10–86		14–146			
AST(IU/L)						
Median	32.5		51.5		−3.553	<0.001 **
Range	8–88		24–107			
Albumin (g/dL)						
X ± SD	4.14 ± 0.37		4.06 ± 0.48		1.04	0.301
Range	3.4–4.8		3.1–5.0			
Creatinine (mg/dL)						
X ± SD	0.85 ± 0.26		0.75 ± 0.21		2.266	0.026 *
Range	0.6–1.3		0.5–1.4			
AFP					Z	
Median	3.8		5.55		−3.252	0.001 **
Range	1.7–16		1.3–149.5			
PCR level of HCV RNA before treatment (IU/mL)						
Median	1.48 × 10^6^		1.72 × 10^6^		Z	
Range	(0.4–6.2) × 10^6^		(0.002–7.9) × 10^6^		−1.077	0.077

t: independent sample *t*-test; χ^2^: Chi square test; Z: Mann Whitney test. BMI: body mass index; WBCs: white blood cells; PLT: platelets count; INR: the international normalized ratio of prothrombin time; ALT: serum alanine aminotransferase; AST: serum aspartate aminotransferase; AFP: alpha-fetoprotein. * *p* < 0.05: statistically significant ** *p* ≤ 0.001: highly significant statistically.

**Table 2 tropicalmed-07-00230-t002:** Ultrasound (US) findings of the liver, FIB4, and treatment protocol among studied groups.

	Responder Group I		Non-Responder Group II		χ^2^	*p*
	*n*= 54	%	*n*= 54	%		
Normal	6	11.1	14	25.9	26.656	<0.001 **
Parenchymatous liver	44	81.5	18	33.3		
Cirrhotic liver	4	7.4	22	40.7 *		
FIB4 < 1.45 = f0–f1 (METAVIR < f2 none to moderate fibrosis)	41	75.9	31	57.4	4.167	0.041 *
FIB4 ≥ 1.45 (METAVIR ≥ f2 Significant fibrosis or cirrhosis).	13	24.1	23	42.6		
SOF/DCV SOF/DCV/RBV	34 20	63.0 37.0	24 30	44.4 55.6	3.724	0.054

χ^2^: Chi square test. * *p* < 0.05: statistically significant ** *p* ≤ 0.001: highly significant statistically.

**Table 3 tropicalmed-07-00230-t003:** Genotype and allele frequencies *IL28B (rs12979860)* distribution through the responder and non-responder patients.

	Total Patients *n*= 108 (%)	Responders Group I		Non Responders Group II		COR (95% CI)	*p*
		*n* = 54	%	*n* = 54	%		
Genotype N (%)							
CC	38 (35.19)	28	51.9	10	18.5	1 (reference)	
CT	30 (27.77)	12	22.2	18	33.3	4.2 (1.5–11.73)	0.007 *
TT	40 (37.04)	14	25.9	26	48.1	5.2 (1.97–13.74)	<0.001 **
Allele, n (%)							
C	106 (49.1)	68	63	38	35.2	3.13 (1.08–5.64)	<0.001 **
T	110 (50.9)	40	37	70	64.8		

* *p* < 0.05: statistically significant; ** *p* ≤ 0.001: highly significant statistically; COR: Crude odds ratio; CI: confidence interval.

**Table 4 tropicalmed-07-00230-t004:** The relation between *IL28B (rs12979860)* genotype, demographics, and laboratory parameters.

	CC *n*= 38	CT *n*= 30	TT *n*= 40	F	P
Age	46.53 ± 12.83	51.5 ± 10.36	53.65 ± 8.2	4.559	0.013 *
HSD	P1 0.138	P2 0.68	P3 0.01 *		
BMI	30.26 ± 3.26	30.45 ± 4.78	31.74 ± 6.53	0.968	0.383
WBC (103/mm^3^)	6.43 ± 1.41	6.61 ± 2.15	6.51 ± 1.94	0.084	0.92
Hb (g/dL)	13.76 ± 1.81	13.28 ± 1.41	13.04 ± 1.34	2.231	0.112
PLT (103/mm^3^)	201.68 ± 48.65	209.17 ± 86.58	173.43 ± 67.54	2.83	0.064
INR	1.13 ± 0.2	1.23 ± 0.12	1.18 ± 0.21	1.23	0.279
T. bil (mg/dL)	0.73 ± 0.14	0.92 ± 0.32	0.86 ± 0.26	5.08	0.008 *
HSD	P1 0.008 *	P2 0.575	P3 0.071		
Albumin (g/dL)	4.18 ± 0.31	4.1 ± 0.42	4.02 ± 0.51	1.325	0.27
Creat. (mg/dL)	0.74 ± 0.15	0.88 ± 0.28	0.79 ± 0.26	3.109	0.049 *
HSD	P1 0.038 *	P2 0.54	P3 0.285		
	Median (range)	Median (range)	Median (range)	KW	p
ALT (IU/L)	40.5 (13–146)	40.1 (24–115)	38 (10–70)	2.435	0.296
AST (IU/L)	35 (15–88)	36 (23–107)	36 (8–78)	1.309	0.52
FIB-4	1.35 (0.2–3.6)	1.8 (0.75–7.4)	3.06 (0.6–9.38)	11.963	0.003 *
Pairwise	P1 0.009 *	P2 > 0.999	P3 0.009 *		
AFP	3.6 (1.7–9.2)	7.75 (1.3–149.5)	4.45 (1.9–23.5)	12.751	0.002 *
Pairwise	P1 0.001 **	P2 0.522	P3 0.062		
PCR level of HCV RNA before treatment (IU/mL)	1.4 × 10^6^ (0.68–7.9) × 10^6^	2.5 × 10^6^ (0.44–5.9) × 10^6^	2.4 × 10^6^ (0.002–6.2) × 10^6^	11.758	0.003 *
Pairwise	P1 0.132	P2 0.75	P3 0.002 *		

F: One way ANOVA test; KW: Kruskal Wallis test; HSD: Tukey highest significance test; * *p* < 0.05: statistically significant; ** *p* ≤ 0.001: highly significant statistically; p1: the difference between CC and CT genotype; P2: the difference between CT and TT genotype; p3: the difference between CC and TT genotype.

**Table 5 tropicalmed-07-00230-t005:** The relation between *IL28B (rs12979860)* genotype, ultrasound findings of the liver, and treatment protocol.

	CC *n*= 38		CT *n*= 30		TT *n*= 40		χ^2^	*p*
	*n*.	%	*n*.	%	*n*.	%		
US								
Normal	10	26.3	0	0.0	10	25		
PLD	26	68.4	21	70	15	37.5	21.12	<0.001 **
Cirrhotic liver	2	5.3	9	30	15	37.5		
*p*	*p**1* < 0.001 **		*p**2* 0.281		*p**3* 0.025 *			
Treatment								
SOF + DCV	32	84.2	11	36.7	15	37.5	21.95	<0.001 **
SOF + DCV + RBV	6	15.8	19	63.3	25	62.5		
P*p*	*p1* < 0.001 **		*p2* 0.86		*p3* < 0.001 **			

χ^2^: Chi square test: ** *p* ≤ 0.001: statistically highly significant; * *p* < 0.05: statistically significant; *p1*: the difference between CC and CT genotype; *p2*: the difference between CT and TT genotype; *p3*: the difference between CC and TT genotype.

**Table 6 tropicalmed-07-00230-t006:** Binary logistic regression analysis of the factors associated with non-response.

	β	*p*	AOR	95% C.I.	
				Lower	Upper
ALT	0.044	<0.001 **	1.044	1.02	1.07
CC genotype		0.002 *			
CT genotype	2.171	0.004 *	8.768	2.022	38.023
TT genotype	2.338	0.001 **	10.364	1.748	28.525
Normal US		<0.001 **			
Parenchymatous liver disease	−1.617	0.022 *	0.198	0.05	0.789
Cirrhosis	0.88	0.302	2.412	0.454	12.817

* *p* < 0.05: statistically significant; ** *p* ≤ 0.001: statistically highly significant; AOR: adjusted odds ratio; CI: Confidence interval.

## Data Availability

The data are available upon reasonable request from the corresponding author and Appendix A.

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
