# Peer review of "Interleukin 28B Polymorphism as a Predictor of Sustained Virological Response to Sofosbuvir-Based Therapy for Hepatitis C Virus Patients"

_tropicalmed, 2022, doi:10.3390/tropicalmed7090230_

Round 1
Reviewer 1 Report
Thank you very much for inviting me to review the manuscript entitled “Interleukin 28B polymorphism as a predictor of sustained viro”.
Manuscript is acceptable for publication after minor correction as bellow:
1. The use of the word “Egypt” in the title is not appropriate because it does not cover the patients of the entire country of Egypt and probably covers a region or a city.
2. MS need to revise typing errors.
3. It is suggested to draw CONSORT flow diagram of patient enrolled through the clinical.
4. New references are not used.
5. Considering doing similar studies, the most important point for novelty should be stated.
Reviewer 2 Report
The rate of non-responder is relatively high (around 50%). Would authors describe it and include in the manuscript.
What was the genotype of HCV in this study? Please include and compare in responder and non-responder groups.
What was the mean level of HCV RNA load in non-responder group? Dose it associated with host and viral parameter (genotype, AST, ALT and so on.....)
Statistical analysis is so long, needs to be shortage.
Limitation of the study such as low number of subjects should be include in the study
Reviewer 3 Report
Methods
1) The authors need to show how the sample size come out with n=108. Any references fr this?
2) Sampling - Please describe DNA extraction method & Genotyping method in sentences form.
3) Discussion & Conclusion - need to improve
Reviewer 4 Report
The manuscript entitled "Interleukin 28B polymorphism as a predictor of sustained virological response to sofosbuvir- based therapy for HCV in Egyptian patients" written by Zaki et al. is of interest to the community. Given the exceedingly high prevalence of HCV in the Egyptian population, a greater understanding of factors associated with attaining SVR is of critical importance.
The study design and data gathered from the study are of sound methodology. The concern would be increasing the quality of the presentation of the work and further explaining the significance of its findings.
A more in depth introduction would serve to clearly frame the purpose of the study. In the introductions current form, it doesn't clearly cover the depth of IL28B SNPs in HCV treatments and what is currently known save for a single reference.
Although the findings of the study itself are of interest, greater explanation of why they are of interest would be needed. For instance, the finding that "Moreover, a statistically substantial relationship was observed between IL28B (rs12979860) SNP and age (p = 0.013), " was observed is important in the fact that there may be an overrepresentation of an age group in a particular cohort that should be considered when evaluating the findings is very important. However, this point is never made and simply points out the statistical significance of the relationship (which is less impactful). A similar concern are the conclusions that may potentially be circular, such as the AST liver enzyme levels and presence of cirrhosis. As it is stated currently, "Cirrhosis increases the risk of non-response by 2.412-folds " which is a very interesting finding, however there remains the potential that the cirrhosis and elevated AST is resultant from the failure of the patient to achieve SVR and therefore a consequence of that rather than a predictive quality of treatment efficacy. Although one cannot ascertain which is more accurate, as it is a retrospective cohort, addressing this possibility would be beneficial.
The discussion is a bit difficult to digest as it is simply a list of how the findings of the paper are in agreement or not for other published papers. It feels more as a list rather than presenting how this study fits into the whole of the field and broadly what is the likely consensus of these many studies.
Round 2
Reviewer 4 Report
The manuscript is highly improved and addressed all the comments and concerns of the original submission. The additional information was very clarifying on the findings and increased the clarity of the manuscript as a whole. Great work!